# Exploring the Roles of lncRNAs in GBM Pathophysiology and Their Therapeutic Potential

**DOI:** 10.3390/cells9112369

**Published:** 2020-10-28

**Authors:** Christian T. Stackhouse, G. Yancey Gillespie, Christopher D. Willey

**Affiliations:** 1Department of Neurosurgery, University of Alabama at Birmingham, Birmingham, AL 35233, USA; ctstackh@gmail.com (C.T.S.); gygillespie@uabmc.edu (G.Y.G.); 2Department of Radiation Oncology, University of Alabama at Birmingham, Birmingham, AL 35233, USA

**Keywords:** long non-coding RNA, non-coding RNA, glioblastoma multiforme, glioma, DNA repair, epigenetics, cancer

## Abstract

Glioblastoma (GBM) remains the most devastating primary central nervous system malignancy with a median survival of around 15 months. The past decades of research have not yielded significant advancements in the treatment of GBM. In that same time, a novel class of molecules, long non-coding RNAs (lncRNAs), has been found to play a multitude of roles in cancer and normal biology. The increased accessibility of next generation sequencing technologies and the advent of lncRNA-specific microarrays have facilitated the study of lncRNA etiology. Molecular and computational methods can be applied to predict lncRNA function. LncRNAs can serve as molecular decoys, scaffolds, super-enhancers, or repressors. These molecules can serve as phenotypic switches for GBM cells at the expression and/or epigenetic levels. LncRNAs can affect stemness/differentiation, proliferation, invasion, survival, DNA damage response, and chromatin dynamics. Aberrant expression of these transcripts may facilitate therapy resistance, leading to tumor recurrence. LncRNAs could serve as novel theragnostic or prognostic biomarkers in GBM and other cancers. RNA-based therapeutics may also be employed to target lncRNAs as a novel route of treatment for primary or recurrent GBM. In this review, we explore the roles of lncRNAs in GBM pathophysiology and posit their novel therapeutic potential for GBM.

## 1. Introduction

The vast majority of the human genome is transcribed, however, less than 3% of all transcribed genes are protein-coding. For decades, speculation over the roles of non-coding RNA (ncRNA) has ranged from junk transcripts to master epigenetic regulators. The majority of prior ncRNA research has emphasized the roles of small ncRNAs such as micro RNAs (miRNA). A particular class of ncRNAs, long non-coding RNAs (lncRNAs), have been increasingly recognized as playing large regulatory roles in normal function and in disease etiology [1,2,3,4]. LncRNAs are versatile transcripts that play important roles in subcellular localization, transcriptional regulation, and epigenetic remodeling. As such, they can have broad phenotypic influences. There has been an exponential increase in the number of lncRNA publications in the past decade with more than 90% of all lncRNA publications coming since 2010. This is an area of research that is rapidly evolving and there are many opportunities for breakthrough discoveries across multiple disciplines.

Gliomas are tumors originating from glial cells in the central nervous system (CNS). The most aggressive form of gliomas is glioblastoma (GBM), the most common primary CNS malignancy in adults. GBMs can either be primary or arise from low grade glioma (LGG) with drastic differences in etiology dependent on primary or secondary malignancy. Secondary GBM tend to be IDH1 mutants with or without 1p/19q co-deletion [5]. Primary GBM tend to be IDH1 WT, have greater diversity, and bear poorer prognosis. The current standard of care for GBM is maximal safe surgical resection followed by adjuvant temozolomide (TMZ) chemotherapy and radiation therapy [6]. With the standard of care, median survival is roughly 15 months [7,8]. GBMs are almost universally fatal and invariably recur following therapy [9]. The roles of lncRNAs in GBM have gone unstudied until very recently, and now there is evidence that these transcripts may have prognostic or even therapeutic applications. In this review, we explore what is known about lncRNA function and how it might be applied to future therapeutic strategies in the treatment of GBM.

## 2. Categorization and Structure of lncRNAs

LncRNAs are characterized primarily by length (>200 nucleotides) and lack of coding potential. In some cases, however, lncRNAs can contain short open reading frames for small peptides [10]. LncRNAs can be poly-adenylated, spliced, and 5′ 7-methyl guanosine capped. They also can be unprocessed. This can make lncRNA transcripts difficult to detect in standard poly-A selected sequencing libraries. Adding to the difficulty, lncRNAs exhibit roughly 10× lower abundance in cells than coding genes. These transcripts can be exceptionally long (up to 400kb) such as in the case of asynchronously replicating autosomal RNAs (ASARs) [11]. These transcripts often contain secondary and tertiary structures. Often, lncRNAs contain 3′ or 5′ hairpin structures that aid in transcript stability and can serve as docking sites for enzymes or other proteins. In some cases, lncRNAs are circularized, greatly increasing their stability. The greater stability of lncRNA transcripts often means they can play a greater physiological role despite their lower abundance. The higher order, complex structures of RNAs confer capacities for protein, DNA, or RNA interactions. LncRNAs functions can broadly be divided into 3 categories: sponges, scaffolds, or signals (Figure 1).

## 3. LncRNA Molecular Functions

The diversity of lncRNAs in form lead to a vast variety of molecular functional roles in normal and aberrant biology. Owing to the size of the non-coding transcriptome, a complex regulatory system governing lncRNA expression has evolved. Developmental regulation of lncRNAs mean that lncRNA expression is cell-specific and temporally regulated. Tissue-specific expression of lncRNAs can contribute to the maintenance of differentiation and phenotype. Many EZH2-associated lncRNAs have been shown to have tissue-specific binding potential suggesting a tissue-specific role for lncRNAs in chromatin remodeling and cellular imprinting [12]. This can be accomplished by regulation of transcriptional programs globally through chromatin remodeling or transcriptional repression/activation. Temporal regulation of lncRNA expression may serve to guide differentiation and cell fate again as global regulators of transcription directly or epigenetically. In addition to temporal regulation, lncRNAs can contain localization signals, specifically in their 3′ UTRs which lead to spatial regulation or subcellular localization of lncRNA transcripts. This subcellular localization of transcripts is seen particularly in the nervous system, where cell morphology is complex [13]. Transcriptional regulation by lncRNAs can be accomplished on multiple levels (Figure 1). LncRNAs can bind DNA and recruit proteins to initiate DNA looping resulting in super-enhancer activity either in cis or in trans. UCA1 and HCCL5 are lncRNAs associated with super-enhancer activation of YAP genes and epithelial-to-mesenchymal transition in ovarian and hepatocellular carcinoma respectively [14,15]. The DNA binding potential of lncRNAs also means they can bind to promoter proximal regions and can directly or indirectly be repressive or activating. LncRNAs can recruit transcriptional machinery or DNA methyl-transferases (DNMTs). Antisense binding of lncRNAs can be directly suppressive of transcription by blocking transcriptional machinery from binding. LncRNA transcripts can act as molecular sponges, sequestering transcription factors or miRNAs. As endogenous competing RNAs, these transcripts can either titrate proteins/enzymes, or small RNA transcripts to inhibit their normal activity. Alternatively, these transcripts may be utilized to deliver their binding targets to subcellular locations where their action has a physiological response. This could again be seen in neurons where complex regulation distal to the nucleus is required. LncRNAs can affect mRNA stability either through the sequestration of miRNAs or by binding them directly. Direct RNA binding can lead to subcellular localization, enhanced transcript stability, suppression, or alternative processing. LncRNAs may direct differential splicing in other transcripts or contribute to post-transcriptional modifications. The complex structures and size of lncRNA transcripts make them ideal subunits of ribonucleoprotein (RNP) complexes. LncRNAs play both a structural role in RNP complexes and a functional role in guiding these complexes to specific binding sites. RNPs formed by lncRNAs include interactions with polycomb repressive complex, histone deacetylases (HDACs), histone acetyl transferases, (HATs), etc. In this way, lncRNAs are important contributors to chromatin stability and heterochromatin or euchromatin formation. One important example of broad scale chromatin remodeling is cellular imprinting via the lncRNA Xist, which is responsible for X-chromosome inactivation in females. This is one example of sex-specific lncRNA expression [16]. It is now believed that some lncRNAs may lead to heritable effects through epigenetic modifications [17].

## 4. LncRNAs in Cancer

The versatility and persistence of lncRNAs lead to dramatic phenotypic control over cells. When applied in the context of malignancy, lncRNAs can begin to appear as master regulators of tumorigenic potential. Many lncRNAs have regulatory potential encompassing multiple intracellular or phenotypic processes. LncRNA activity can be seen across the spectrum of canonical cancer hallmarks (Figure 2). However, there are very few mechanistic studies of lncRNAs in the disease context of GBM. In this section, we highlight the known roles of lncRNAs in all cancers to highlight the untapped potential that additional study of lncRNAs in the context of GBM may yield.

### 4.1. Proliferation

Sustained proliferation and avoidance of growth suppressors is one of the key hallmarks of malignant cells. The lncRNA CASC11 has been shown to promote proliferation in bladder cancer through binding of miR-150 [18]. The lncRNA CCAT1, named for its association with colorectal cancer (CRC), has not only been found to promote proliferation in bladder cancer, but also promotes migration and invasion [19]. Also in bladder cancer, the lncRNA GClnc1 promotes proliferation as well as invasion through the activation of proto-oncogene MYC [20]. In cervical cancer, antisense lncRNA DLG1-AS1 promotes proliferation through titration of miR-107 leading to upregulation of ZHX1 [21]. A relatively well studied lncRNA transcript Nuclear Enriched Abundant Transcript 1 (NEAT1) has been found to promote proliferation and epithelial to mesenchymal transition (EMT) pathways in breast cancer [22]. Also in breast cancer, lncRNA SNHG6 plays a role in proliferation and invasion via the miR-26a/VASP signaling axis [23]. Similarly, lncRNA PSMG3-AS1 functions to titrate miR-143-3p leading to increased proliferation and migration in breast cancer [24]. The downregulation of lncRNA TUG1 by another lncRNA GATA6-AS results in enhanced proliferation and inhibition of apoptosis in glioma [25].

### 4.2. Immune Evasion

Cancer cells frequently develop immune evasion mechanisms or even hijack normal immune cells. Tumors can induce pro-tumor inflammatory responses. LncRNAs have been shown to play a role in cancer cell programming related to immunity. The lncRNA SATB2-AS1 is normally expressed in colorectal tissues and is downregulated in CRC leading to an inverse relationship between expression and prognosis. SATB2-AS1 mediates H3 lysine 4 tri-methylation (H3K4me3) deposition and promoter methylation of SATB2, regulating TH1-type chemokine expression and immune cell density [26]. A novel lncRNA LINK-A has been found to suppress tumor antigenicity and immune–related tumor suppression through signaling related to TRIM71 in triple-negative breast cancer (TNBC) [27]. NF-κB-interacting lncRNA (NKILA) has been shown to promote activation-induced cell death in cytotoxic T lymphocytes resulting in immune evasion in breast and lung cancer [28]. Also in breast cancer, SNHG1 mediates the differentiation state of T regulatory cells affecting immune escape via the miR-448/IDO axis [29]. Tumors can also recruit immune cells such as tumor-associated macrophages (TAMs) to aid in their growth and survival. HIF-1α-stabilizing lncRNA (HISLA) is released from TAMs in extracellular vesicles, promoting glycolysis and survival in breast cancer [30]. The lncRNA DGCR5 is related to immune-related biological processes and simultaneously strongly negatively correlated with WHO grade of malignancy in glioma suggesting a role in inflammatory activities or immune checkpoints in glioma [31].

### 4.3. Metastasis/Tumorigenesis

Many cancers possess metastatic potential, typically developed in a subpopulation of perivascular cells. Metastatic cells must migrate into the blood stream of lymphatic system, exit, and establish a new tumor at a distal site. There is evidence that lncRNAs may confer metastatic potential in these highly specialized cells. The novel lncRNA XLOC_006390 has been shown to facilitate tumorigenesis and metastasis in cervical cancer cells by serving as molecular sponges for miR-331-30 and miR-338-3p [32]. LncRNA BX111 is induced under hypoxic conditions and promotes metastasis in pancreatic cancer by activating epithelia-mesenchymal transition (EMT) via induction of ZEB1 transcription [33]. The lncRNA PTAR also promotes EMT and metastasis in ovarian cancer by disinhibiting ZEB1 expression by competitively binding miR-101-3p [34]. The roles of lncRNAs are not always clear cut. MALAT1 has been described as conferring metastatic potential, particularly in lung to brain metastasis through EMT regulation [35,36]. However, in breast cancer, MALAT1 has been found to inactivate pro-metastatic transcription factor TEAD, reducing metastatic potential in these cells [37]. LINC01133 also inhibits breast cancer metastasis through EZH2-mediated regulation of SOX4 expression [38]. Meanwhile, LINC02273 drives metastasis in breast cancer by stabilizing hnRNPL, resulting in activation of AGR2 transcription [39].

### 4.4. Replicative Immortality

Cancer cells characteristically undergo uncontrolled cell division. A hallmark of many cancer cells is their ability to overcome replicative senescence and acquire replicative immortality via enhanced stability of chromosomes often through telomere lengthening [40]. The primary lncRNA related to telomere maintenance is telomeric repeat-containing RNA (TERRA) [41]. TERRA expression has been associated with elongated telomeres in human placenta [42]. Elevated expression of TERRA has been detected in various human cancer cell lines [43]. SENEBLOC is a lncRNA which suppresses senescence through p53 related mechanisms [44]. Similarly, Linc-ASEN represses senescence by reducing p21 expression allowing for uncontrolled cell cycle progression [45]. 

### 4.5. Invasion/Migration

Related to but separate from metastasis, cancer cell migration and stromal invasion have major implications in the progression and treatment of disease. GBM for example, is considered a whole organ disease as tumor cells are highly invasive/migratory and yet, GBM metastasis outside of the CNS is exceedingly rare [46]. Highly infiltrative tumors complicate surgical resection and can lead to disease recurrence. Many lncRNAs are associated with invasive/migratory potential of cancer cells, both promoting and inhibiting. Expression of lncRNA RHPN1-AS1 promotes invasion and migration in cervical cancer by modulating the miR-299-3p/FGF2 axis [47]. Overexpression of PTCSC3on the other hand, inhibits invasion and migration of cervical cancer cells by sequestering miR-574-5p [48]. NEAT1 which as noted earlier promotes EMT and proliferation in breast cancer, additionally promotes migration and invasion in endometrial cancer through regulation of the miR-144-3p/EZH2 axis [49]. In colon cancer, NEAT1 also promotes migration and invasion, in this case through the miR-185-5p/IGF2 axis [50]. Meanwhile, the expression of lncRNAs EZR-AS1, LINC00261, and LINC01082 all suppress migration in colon cancer cells through various signaling pathways [51,52,53]. In glioma, EGFR-AS1 is associated with migration and invasion through the mir-133b/RACK1 axis [54].

### 4.6. Angiogenesis

Solid tumors have increased need for gas and nutrient exchange owing to their rapid growth. Tumor-associated blood vessels are also important in metastasis. Cancer cells frequently have the capacity to promote angiogenesis or can even differentiate into endothelial cells to form new vessels [55]. A novel lncRNA HIF-1α inhibitor at translation level (HITT) is frequently downregulated in human cancers, resulting in increased angiogenesis and tumor growth via HIF-1α expression [56]. HITT accomplishes this repression by coordinating with EZH2 to epigenetically suppress HIF-1α [57]. Another hypoxia-induced lncRNA RAB11B-AS1 promotes angiogenesis and metastasis in breast cancer [58]. The lncRNA NR2F1-AS1 promotes angiogenesis in breast cancer through IGF-1/ERK signaling regulation [59]. NKILA which regulates cancer-immune interactions also regulates angiogenesis in breast cancer through NF-κB/IL-6 signaling pathway modulation [60]. LINC00284 inhibits MEST expression in an NF-κB dependent manner leading to increased angiogenesis in ovarian cancer cells [61]. Differentiation antagonizing non-protein-coding RNA (DANCR) also promotes tumor angiogenesis through the miR-145/VEGF axis in ovarian cancer [62]. The lncRNA H19 promotes angiogenesis in glioma via the miR-138/HIF1α/VEGF axis [63].

### 4.7. Genome Instability and Mutation

Cancer cells classically acquire very high mutation loads owing to their numerous cellular divisions and to genomic instability. This instability can lead to the deletion of tumor suppressors or the amplification of oncogenes. One example of this is PTEN loss in GBM. LncRNA transcripts are emerging as regulators of genomic instability. We have already mentioned how the lncRNA TERRA is associated with telomeres and replicative immortality, but through its roles in chromosome binding, TERRA also plays a role in genomic stability [41]. Another lncRNA associated with cellular division and genomic stability is MANCR (LINC00704), which is upregulated in highly mitotic cells in TNBC [64]. Noncoding RNA activated by DNA damage (NORAD) is a lncRNA which acts as a scaffold for the assembly of topoisomerase complexes critical for maintaining genomic stability [65]. NORAD also promotes genomic stability through the sequestration of destabilizing PUMILIO proteins [66]. Loss of NORAD in some cancers could lead to genomic instability and higher copy number variants (CNV) or mutational loads. Colon cancer associated transcript 2 (CCAT2) has been found to induce chromosomal instability through Bop1 and AURKB signaling pathways [67].

### 4.8. Resisting Cell Death

A hallmark of cancer cells related to replicative immortality, is their ability to avoid regulated, programmed cell death mechanisms. Several lncRNAs have been associated with this hallmark both positively and negatively regulating apoptosis. The well-characterized HOTAIR lncRNA has been found to suppress apoptosis in breast cancer cells through the miR-20a-5p/HMGA2 axis [68]. Similarly, PART1 influences apoptosis in prostate cancer cells through modulation of toll-like receptor pathways [69]. Conversely, the lncRNA ATB promotes apoptosis in non-small cell lung cancer (NSCLC) via the regulatory axis of miR-200a and β-catenin [70]. Loss of the lncRNA MEG3 in prostate cancer cells leads to decreased apoptosis rates through disinhibition of QKI-5 expression via miR-9-5p [71]. LncKLHDC7B has found to be associated with resistance to cell death and poorer prognosis in TNBC [72]. Furthermore, CCAT1, which is associated with cancer progression in CRC and bladder cancer, also regulates apoptosis in U87 glioma cells through sponging of miR-181b [73].

### 4.9. Altered Metabolism

Aberrant metabolism in cancer cells has long been an attractive therapeutic target. Cancer cells frequently undergo the Warburg effect which describes increased energy production through aerobic glycolysis [74]. LINC00504 has been found to promote aerobic glycolysis in ovarian cancer cells through the titration of miR-1244 [75]. The proto-oncogene MALAT1 has been shown in hepatocellular carcinoma to simultaneously increase glycolysis and decrease gluconeogenesis through mTOR-mediated regulation of TCF7L2 expression [76]. The lncRNA EPB41L4A-AS1 serves as a repressor of the Warburg effect in multiple cancers through interaction and co-localization with HDAC2 and NPM1 [77]. In glioma, the lncRNA SNHG14 is downregulated and destabilized owing to a loss in Lin28A, leading to an increase in IRF6-mediated aerobic glycolysis [78]. This suggests that stabilizing SNHG14 could target aberrant metabolism in glioma cells. In contrast, the lncRNA ANXA2P2 is overexpressed in glioma leading to elevated levels of GLUT1, H2K, PFK, and LDHA resulting in increased aerobic glycolysis, suggesting targeting of this lncRNA transcript could be an effective therapeutic strategy [79].

### 4.10. Stemness/Multipotency

Cancer stem cells (CSCs) are believed to be responsible for tumorigenesis and therapy resistance. CSCs have the capacity for self-renewal, multipotency, and survival. The replication and differentiation of CSCs create a hierarchical cell model within tumors, contributing to their malignancy and resilience. Niches within the tumor microenvironment (TME) are thought to support CSCs, but it has been discovered that lncRNAs contribute to the maintenance of stemness and of cellular fate programs. A usual suspect, MALAT1 has been shown to promote stemness in gastric cancer cells by enhancing SOX2 mRNA stability [80]. In a potential feed-forward mechanism, it has been found that the stemness-related transcription factor, Oct4 promotes the expression of MALAT1 and NEAT1 [81]. NEAT1 also confers stem-like phenotypes in TNBC, NSCLC, and GBM cells [82,83,84]. The YAP transcription factor is a potent oncogene related to several oncogenic programs including stemness [85]. The lncRNA B4GALT1-AS1 serves to recruit YAP to the nucleus thus enhancing its transcriptional activity and enhancing stemness in colon cancer cells [86]. In glioma, the lncRNA SNHG20 has been shown to promote stemness through activation of the PI3K/Akt/mTOR signaling pathway [87].

### 4.11. DNA Damage Response

The capacity of cancer cells to overexpress DNA repair machinery is an essential component of their resistance to conventional therapies including ionizing radiation or alkylating chemotherapy. Some lncRNAs regulate DNA damage response (DDR) transcriptional programs. NEAT1 induces expression of multiple DDR pathways including homologous recombination (HR) in multiple myeloma (MM) [88]. NEAT1 also serves as a key structural component in paraspeckle formation which may in turn play a role in DDR regulation and activity [89]. The lncRNA in NHEJ pathway 1 (LINP1) has been found to promote PARP-dependent DNA repair in TNBC as a structural component of the IGFBP-3/NONO/SFPQ complex [90]. Alternative NHEJ in MM may also be mediated by MALAT1 through its interaction with PARP1 and LIG3 [91]. MALAT1 has also been shown to be highly upregulated in TMZ-resistant GBM cells and that the knockdown of MALAT1 transcripts rescues TMZ sensitivity by way of increasing miR-101 [92].

### 4.12. Recruitment of Stromal Cells

Tumors, especially highly infiltrative tumors like gliomas, often encompass normal, non-malignant cells in their mass. Cancer cells are often able to co-opt normal cells to support their TME and facilitate growth/spread. Immune cells, particularly TAMs, can not only contribute to immune evasion, but also to growth. HISLA is released in extracellular vesicles from TAMs resulting in enhanced aerobic glycolysis and apoptotic resistance in nearby breast cancer cells [30]. Carcinoma-associated fibroblasts (CAFs) in CRC transfer lncRNA H19 via exosomes resulting in enhanced stemness and chemoresistance [93]. A novel lncRNA lnc-CAF, so named because of its overexpression in CAFs, promotes cancer cell growth in oral squamous cell carcinoma (OSCC) through its interaction with interleukin-33 [94]. Hyaluronan is a key ECM component in the CNS and thus very important in glioma progression. An enzyme key to TME maintenance is Hyaluronan synthase 2 (HAS2). The antisense lncRNA HAS2-AS1 has been shown in some cancers to stabilize or enhance HAS2 expression while being repressive in others [95]. CREB1 induces HAS2-AS1 expression in OSCC leading to enhanced proliferation and invasion through the miR-466/RUNX2 axis [96]. A novel lncRNA activated by TGF-β (lncRNA-ATB) is released in exosomes by normal astrocytes and promotes glioma cell invasion [97].

## 5. LncRNAs in Glioma and GBM

The roles of lncRNAs in glioma etiology is an emerging field. Many of the hallmark characteristics of glioma tumors have been found to be regulated by lncRNAs. The lncRNA regulatory story in glioma starts with tumor initiation and progression. A set of p53-regulated lncRNAs initially identified in colon cancer, were found to be expressed inversely with glioma tumor grade with lowest expression in GBM samples [98]. These lncRNAs regulate SOX factor expression in an inverse manner leading to higher SOX expression in GBM tumors with low p53-regulated lncRNA expression. Cancer susceptibility candidate 7 (CASC7) is another lncRNA which appears to inhibit glioma formation and progression by decreasing Wnt/β-catenin signaling activity [99]. On the other hand, CASC9, miR-519d, and STAT3 form a positive feedback loop promoting glioma formation and tumorigenesis [100]. The lncRNA AGAP2-AS1 promotes glioma initiation by sponging microRNAs resulting in activation of the Wnt/β-catenin signaling pathway [101]. The oncogenic lncRNA NEAT1 also regulates tumor initiation and progression via Wnt/β-catenin regulation in an EGFR-dependent manner through its interaction with polycomb repressive complex subunit EZH2 [102]. The PI3K/Akt signaling axis is another important pathway in glioma formation. LINC01426 promotes this signaling and thus glioma initiation [103]. Meanwhile the lncRNA PART1 acts as a tumor suppressor by sponging miR-190a-3p leading to downregulation of PTEN/Akt signaling [104].

The roles of lncRNAs and their mechanisms of action are diverse. Many lncRNAs act on pathways in GBM via regulation of miRNAs either to promote or suppress glioma progression. LINC01446 promotes disease progression via the miR-489-3p/TPT1 axis [105]. Similarly, the lncRNA MNX1-AS1 promotes GBM progression through the inhibition of miR-4443 [106]. The lncRNA DCST1-AS1 decreases mir-29b levels through methylation leading to the promotion of proliferation [107]. A novel lncRNA transcript AC016405.3 acts as a tumor suppressor by acting as a molecular sponge for miR-19a-5p leading to TET2 modulation [108]. Other lncRNAs affect glioma programs through DNA methylation or chromatin modifications. The lncRNA HOTAIRM1 acts as a scaffold facilitating long-range chromatin interactions with HOXA gene clusters, increasing their transcription resulting in more malignant tumors [109]. Additionally, HOTAIRM1 has been shown to titrate EZH2 and DNMTs blocking repression of the HOXA gene clusters among others [110]. An antisense lncRNA HOXB13-AS1 regulates HOX gene transcription by guiding EZH2 to specifically repress translation of HOXB13 [111]. LINC00467 represses the tumor suppressor p53 in glioma formation through its direct interactions with DNMT1 [112].

One of the primary obstacles in the treatment of glioma are the extent of intra and intertumoral heterogeneity. LncRNA expression has been found to be dynamic across multiple single cells in GBM tumors and cell lines [113,114]. There are three canonical molecular subtypes of GBM cells: classical/proliferative, proneural, and mesenchymal. Generally mesenchymal cells are considered the most malignant and therapy resistant. There is also the concept of stem-like gliomas stem cells (GSCs) which are considered responsible for therapeutic failure and tumor recurrence. Much of the heterogeneity of glioma tumors is dependent upon TME niches. The lncRNA HIFiA-AS2 has been found to maintain mesenchymal GSCs in hypoxic niches within the tumor [115]. As mentioned previously, HOTAIRM1 also contributes to the maintenance of GSCs through the regulation of HOX gene expression. 

Gliomas and GBM are highly infiltrative of normal stroma and are considered a whole organ disease. The invasive and migratory potential of glioma cells contributes to the difficulties in treating this disease and the eventual recurrence of these tumors. Glioma invasion is promoted in a HIF1α-dependent manner through the expression of lncRNAs H19 and AWPPH [63,116]. Migration is also promoted by LINC01494 which titrate miR-122-5p leading to increase expression of CCNG1 [117]. NEAT1 expression increases invasive potential of glioma cells through modulation of SOX2 via miR-132 [118]. The lncRNA ATB promotes glioma cell invasion through NF-κB and MAPK signaling pathways [119]. GAS5 on the other hand is a lncRNA which acts to suppress invasion and tumor growth in glioma cells by targeting GSTM3 expression [120].

Another hallmark of glioma and GBM is acquired resistance to standard therapies. The canonical mechanism of TMZ resistance in GBM is the expression and activity of DNA repair enzyme O6-methylguanine-DNA methyltransferase (MGMT). Lnc-TALC promotes MGMT expression by positively regulating the c-Met pathway [121]. Treatment using alkylating chemotherapy or radiation therapy induces the expression of lncRNAs such as MALAT1 induction in an NF-kB and p53 codependent manner following TMZ treatment [122]. Furthermore, the lncRNA ADAMTs9-AS2 promotes TMZ resistance through changes in ubiquitination mediated by FUS/MDM2 [123]. The lncRNA TP73-AS1 has been found to promote TMZ resistance in GSCs through its regulation of a GSC/therapy resistance marker ALDH1A1 [124]. NCK1-AS1 also increases TMZ resistance through the disinhibition of TRIM24 in its function as a competing endogenous RNA [125]. Radiation therapy is the most common and widely tolerated therapeutic approach in treating glioma. Several lncRNAs have been associated with response to DNA damage following radiotherapy. Using a cohort from the TCGA, 37 lncRNAs were found to be associated with radiosensitivity in LGG mostly related to PI3K-Akt, MAPK signaling, and DDR [126]. The lncRNAs HMMR-AS1 and TALNEC2 have been found to confer radiation resistance which can be rescued by silencing these transcripts [127,128]. TALNEC2 also appears to regulate growth and stemness in glioma stem cells [128]. The antisense transcript of hypoxia-inducible factor-1α (AHIF) has been found to confer radioresistance in GBM cells and can transfer this resistance to neighboring cells through exosome transmission [129]. Knocking down the lncRNA PCAT1 has been found to increase glioma cell sensitivity to radiation, possibly through regulation of transcriptional modifying gene HMGB1 [130].

## 6. LncRNAs as Biomarkers

Much research goes into the discovery and classification of ever elusive disease biomarkers. Biomarkers come in three not always exclusive flavors: diagnostic, prognostic, and theragnostic. Diagnostic biomarkers are useful in distinguishing between normal tissue and cancer. Prognostic indicators exhibit a correlation of expression with prognosis/disease progression independent of therapy. Theragnostic biomarkers have detectable changes in expression which are predictive of therapeutic response. Cancer-related lncRNAs are sometimes detectable in serum and have also been found in circulating exosomes [131]. Serum levels of MEG3, classically a tumor repressor, can be of diagnostic value in CRC [132]. Cancer susceptibility candidate 9 (CASC9) has been found to be upregulated in hepatocellular carcinoma compared to normal healthy controls and levels of expression in HCC may also be predictive of metastasis and prognosis [133]. A number of lncRNAs are induced by radiation therapy [134,135]. LINP1 is induced following radiation in cervical cancer cells to facilitate DNA repair [136]. Many studies have honed in on the systematic identification of lncRNA-based biomarkers in glioma. Differences in serum levels of HOTAIR validated by qRT-PCR between GBM and control patients may represent diagnostic and prognostic biomarkers for GBM [137]. MALAT1 could be a prognostic and/or theragnostic indicator in glioma as its expression is associated with increased chemoresistance to TMZ [138]. A recent review of LGG samples from The Cancer Genome Atlas (TCGA) revealed 16 immune-related lncRNAs strongly correlated with patient prognosis [139]. Also from TCGA, there is a six lncRNA signature related to immunity which is a positive prognostic indicator in GBM [140]. The lncRNA RPSAP52 is expressed in GSCs and expression can be used to predict postoperative survival in GBM patients [141].

## 7. LncRNAs as Therapeutic Targets

The multitude of functions of lncRNAs makes these transcripts attractive potential therapeutic targets as they are often low in abundance yet have broad and significant phenotypic effects. Table 1 shows a list of lncRNAs implicated in GBM along with their roles and potential therapeutic strategies. LncRNA activities are sometimes even conserved across multiple forms of cancer making them potential pan-cancer therapeutic targets. The lncRNA PVT1 is upregulated in a number of cancers including NSCLC, HCC, breast cancer, and glioma [142]. This makes PVT1 and other commonly conserved lncRNAs such as MALAT1, H19, and NEAT1 attractive general therapeutic targets. Sometimes lncRNAs are lost in disease states such as the tumor suppressors MEG3 and GAS5. This could be exploited by drugs that can stabilize or restore the function of tumor suppressor lncRNAs. The lncRNA MATN1-AS1 is consistently down-regulated in GBM and this suppression is associated with poorer prognosis, enhanced proliferation, and increased invasion potential [143]. Upregulation of MATN1-AS1 mitigates this malignant phenotype through inhibition of RELA via E2F6 leading to suppression of MAPK signaling [143]. The clearest path to targeting overexpressed lncRNA is through direct RNA-interference (RNAi). This type of therapy can be accomplished by introducing siRNA, miRNA, shRNA, or even lncRNAs through inducible vectors or nanoparticle delivery systems. A nanoparticle containing siRNAs against MALAT1 was effective at sensitizing GBM cells to TMZ [144]. MALAT1 and its interacting partner AR-v7 have both been targeted successfully in preclinical trials of therapy resistant prostate cancer [145]. Another approach is to target pathways related to lncRNA expression using small molecule inhibitors. Targeting LINK-A transcripts or downstream GPCR signaling with agonists leads to a sensitization of TNBC tumors to immune checkpoint inhibitors [27]. There are even some phytochemicals which are known to modulate common lncRNAs [146]. Melatonin, a simple, BBB penetrant compound has been shown in HCC cells to induce the expression of RAD51-AS1 which blocks the transcription of RAD51, a gene key in HR, leading to increased sensitivity to DSBs from chemo or radiotherapy [147]. The lncRNA HOTAIR is functionally related to many cancer hallmarks including in glioma as previously stated. HOTAIR influences oncogenic programs largely through chromatin remodeling. A novel compound AC1Q3QWB has been discovered which disrupts the HOTAIR-mediated recruitment of polycomb repressive complex 2 leading to sensitization of tumors to other chemotherapeutic agents [148]. 

## 8. Conclusions

LncRNAs are versatile and prolific transcripts in normal biological activity as well as in aberrant disease states. They can exist in many forms with a wide variety of functional capacities. Expression is regulated in a cell-specific manner, as well as spatially and temporally. Despite their low abundance levels, lncRNAs can have major impacts on cellular programs and phenotypes. LncRNAs exhibit roles in all of the canonical hallmarks of cancer with potentials for conserved, pan-cancer activities that may be targetable. These transcripts are particularly expressed in the CNS environment and naturally play a multitude of roles in the most common primary brain malignancy, GBM. GBM remains an intractable, incurable disease for which standard therapies have little effect. These molecules could represent a novel class of biomarkers in GBM or even attractive therapeutic targets. A number of lncRNAs are associated with tumorigeneses, stemness, invasion, and therapy resistance in glioma. We have just begun to scratch the surface of the wide variety of roles of lncRNAs in glioma pathology, particularly in therapy resistance. Further exploration of lncRNAs in glioma may revolutionize our understanding of the disease and may result in novel therapeutic advances in what has up until now been a losing battle.

## Figures and Tables

**Figure 1 cells-09-02369-f001:**
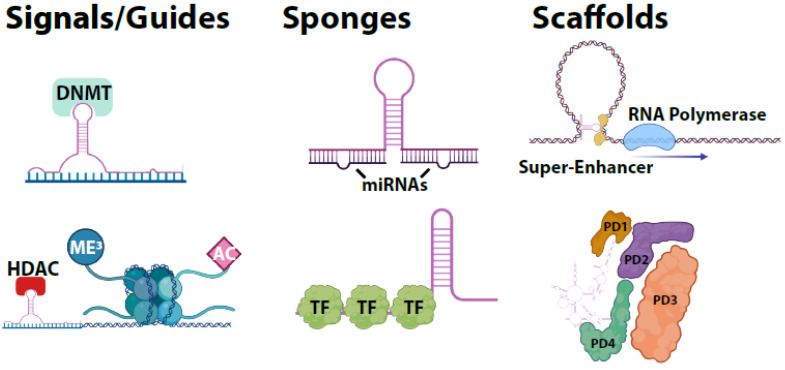
Archetypes of long non-coding RNAs (lncRNAs). LncRNAs can act as signals or guides directing the initiation or repression of transcription. LncRNAs can act as molecular sponges binding miRNAs or transcription factors/proteins. LncRNAs can act as scaffolds to form enhancer loops or as structural components of ribonucleoprotein complexes. DNMT–DNA methyltransferase, HDAC–histone deacetylase, Me–Methyl group, AC–Acetyl group, TF–transcription factor, PD–protein domain. Created with BioRender.com.

**Figure 2 cells-09-02369-f002:**
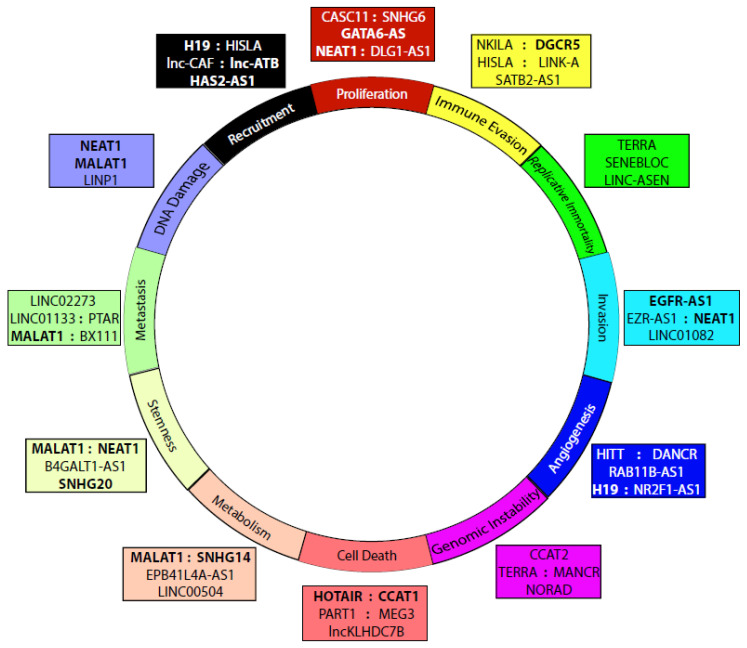
Hallmark characteristics of cancer are regulated by lncRNAs. Colored boxes contain lncRNAs functionally linked to regulation of the corresponding cancer hallmark. Recruitment refers to the recruitment or association with stromal cells. LncRNAs with known association to gliomas are highlighted in bold (Discussed in Section 5).

**Table 1 cells-09-02369-t001:** lncRNAs in glioblastoma (GBM).

lncRNA	Role	Level of Evidence	Therapeutic Strategy	Reference
CASC7	Tumor suppression	GBM primary tissue and cell lines in vitro	Rescue, stabilize, or OE	[99]
CASC9	Tumorigenesis	GBM cell lines in vitro and in vivo	Degrade, KD, or saturate with miRNA mimetic	[100]
AGAP2-AS1	Proliferation and survival	GBM primary tissue, cell lines in vitro, and correlated with OS in TCGA patients	Degrade, KD, saturate with miRNA mimetic, or target downstream Wnt signaling	[101]
NEAT1	Proliferation and invasion	GBM primary tissues, cell lines in vitro and in IC xenograft	Degrade, KD, saturate with miRNA-mimetic, or target SOX2, EGFR, or EZH2	[102,118]
LINC01426	Proliferation, invasion, and survival	TCGA clinical associations, GBM primary tissues, and cell lines in vitro	Degrade, KD, or target PI3K/Akt signaling	[103]
PART1	Tumor suppression and growth inhibition	GBM tissues, cell lines in vitro, and TCGA clinical associations	Rescue, stabilize, OE, or target PI3K/Akt signaling	[104]
LINC01446	Tumorigenesis and progression	Clinical associations, GBM cell lines in vitro and in xenograft	Degrade, KD, OE miR-489-3p, or target TPT1	[105]
MNX1-AS1	Proliferation, migration, and invasion	GBM tissues and cell lines in vitro	Degrade, KD, or OE miR-4443	[106]
DCST1-AS1	Proliferation	Clinical associations, GBM primary tissues and primary culture	Degrade, KD, OE miR-29b, saturate with miR-mimetic	[107]
AC016405.3	Suppression of Proliferation and Invasion	Clinical associations, GBM primary tissues and cell lines in vitro	Rescue, stabilize, OE, sponging/KD miR-19a-5p, or OE of TET2	[108]
HOTAIRM1	Proliferation, invasion, and survival	TCGA clinical associations, GBM tissues, cell lines in vitro and in vivo	Degrade, KD, OE of G9a and EZH2, or KD of HOXA1	[109,110]
HOXB13-AS1	Proliferation/cell cycle progression	GBM tissues, cell lines in vitro and in vivo	Degrade, KD, KD of DNMT3B, OE of HOXB13	[111]
LINC00467	Proliferation and invasion	GBM cell lines (U87, LN229) in vitro	Degrade, KD, target DNMT1, rescue p53 activity/expression	[112]
HIFiA-AS2	GSC maintenance	GBM cell lines in vitro and in vivo	Degrade, KD, target IGF2BP2, DHX9, or HMGA1	[115]
H19	Proliferation, invasion, and angiogenesis	GBM cell lines (HEB, U87, A172, U373) in vitro	Degrade, KD, OE miR-138, target HIF-1α and VEGF	[63]
LINC01494	Proliferation and invasion	Clinical associations, GBM tissues, and cell lines in vitro	Degrade, KD, OE miR-122-5p, target CCNG1	[117]
ATB	Invasion	GBM cell lines (LN-18, U251) in vitro	Degrade, KD, target TGF-β, NF-κB (pyrrolidinedi-thiocarbamate ammonium), and P38/MAPK (SB203580	[119]
GAS5	Inhibition of proliferation, invasion, survival	GBM cell lines (HEB, U251, U87) in vitro	Rescue, stabilize, OE, target GSTM3	[120]
Lnc-TALC	Promotes TMZ resistance and tumor recurrence	TMZ-selected GBM cell lines (LN229, U251, 551W, HG7) in vitro	Degrade, KD, OE miR-20b-3p, target c-Met, AKT/FOXO3, and MGMT	[121]
MALAT1	TMZ resistance and invasion	Clinical associations, GBM patient tissue and serum, GBM cell lines (U87) in vitro and in IC xenograft	Degrade, KD, ASC-J9®, target NF-κB, or restore p53 activity/expression	[122,138,144,145]
ADAMTs9-AS2	TMZ resistance	Clinical associations, GBM cell lines (T98G-R, U118-R) in vitro	Degrade, KD, targeting FUS/MDM2 axis	[123]
TP73-AS1	TMZ resistance and metabolism in GSCs	TCGA clinical associations, GSC lines (G26, G7) in vitro	Degrade, KD, targeting ALDH1A1	[124]
NCK1-AS1	TMZ resistance	GBM patient primary tissue, GBM cell lines (U251, A172) in vitro	Degrade, KD, OE of miR-137, targeting TRIM24	[125]
HMMR-AS1	Tumorigenesis, proliferation, invasion, radiation resistance	GBM cell lines (U87, U251, A172, U118) in vitro	Degrade, KD, target/disrupt HMMR interaction, target ATM, RAD51, BMI1	[127]
TALNEC2	Tumorigenesis and radiation resistance	TCGA clinical associations, GBM primary tissue, GBM cell lines (A172, U251, U87, T98G and LNZ308) in vitro	Degrade, KD, target E2F1	[128]
AHIF	Invasion, survival, radiation resistance	GBM cell lines (U87, U251, A172, T98G) in vitro	Degrade, KD	[129]
PCAT1	Stemness, survival, DNA repair	GBM cell lines in vitro	Degrade, KD, upregulate miR-129-5p, target HMGB1	[130]
HOTAIR	Proliferation, invasion, therapy resistance, chromatin remodeling	Clinical associations, GBM patient tissue/serum, cell lines in vitro and in IC xenografts	Degrade, KD, AC1Q3QWB and DZNep combinational therapy, target EZH2	[137,148]
RPSAP52	Stemness and poor patient prognosis	GBM primary tissue, clinical associations, GBM cell lines (U-373 MG) in vitro	Degrade, KD, target TGF-β1	[141]
MATN1-AS1	Tumor suppressor, suppresses proliferation and invasion	GBM primary tissue and cell lines in vitro and in vivo	Rescue, stabilize, OE, target RELA, ERK1/2, Bcl-2, survivin, or MMP-9	[143]

OE–over express, KD–knock-down, OS–overall survival, TCGA–The Cancer Genome Atlas, IC–intracranial, GSC–glioma stem-like cells.

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
