# Peer review of "Exploring the Roles of lncRNAs in GBM Pathophysiology and Their Therapeutic Potential"

_cells, 2020, doi:10.3390/cells9112369_

Round 1

Reviewer 1 Report

This review is well organized and constitutes an important contribution to the field. The figures are useful and esthetic. A lot of information is presented however, on all cancers. Considering that this review is focused on GBM, I found it harduous sometimes to find the information related to GBM itself. Consider adding a table focusing on lncRNA in GBM, as I suggest below.

Line 39: the origin of GBM from glial or neural stem cells is debated, please remove this

Line 40: replace glial tumors with gliomas

Line 40: remove "multiform", it is no longer used in the WHO terminology

Figure1 : for this figure, more explanation is needed to understand the different elements and their relationships (eg what are the green blobs?). A more obvious color code could be used for the lncRNA so they stand out.

Section 3: concrete examples for these functions would make the text more dynamic

Line 84-87: add reference to figure 1 here

Section 4: This section excludes GBM if I understand correctly. GBM is discussed in Section 5. Why not merge both sections? Or place GBM first? If a reader expects to read about GBM in this review, they might not see that GBM is discussed in the next section.

Figure 2: put one lncRNA per line in the boxes to improve lisibility, since the terminology is variable. Recruitment of what? Resistance to cell death. Add at the end of sentence in the legend: (...) and will be discussed specifically in section 5

Section 5: Would be useful, for all examples listed, to mention in which type of models they were acquired (traditional serum-grown cell line, patient-derived cultures, tissue samples etc). As discussed above, consider merging with section 4. Make sure here that all lncRNA in bold in figure 2 are mentioned here and discussed (I did not check).

Line 371: clarify why HOTAIR could be a diagnostif and prognostic biomarker. In general, indicate why the referred authors reported this conclusion ie was it validated or is it just speculative?

Section 7: again state in which kind of model the findings were made. In general, the examples linked to GBM are lost among the other examples. Overall, and this applies for the entire reviwe, it could be useful to add a table where all the relevant lncRNA in GBM are listed, with references, aspect studied (cellular function, therapeutic target, biomarker), and the type of model

Line 420: yes and the table that I suggested would highlight the need for more research in lncRNA in GBM,

Reviewer 2 Report

The review paper entitled:“Exploring the roles of lncRNAs in GBM pathophysiology and their therapeutic potential”, is overall well written and interesting, where some scientific reports regarding the key roles of lncRNAs on cancer and on the GBM, are summarized.

However, although the authors try to be exhaustive in describing the lncRNAs roles in GBM, in the manuscript there is an imbalance between the roles of lncRNAs reported on other types of cancer and their specific roles described on the GBM. Therefore, in my opinion the topic on the GBM should be improved, and in particular the section 7 (“LncRNAs as Therapeutic Targets”), also considering that this aspect is a part of the manuscript title.

In addition, some minor linguistic changes should be addressed by authors to ensure the manuscript suitable for publication. For example, in line 229 of the text “METABLISM” must be corrected.

The reference list should be reported according to the Journal style (for example, the first author name only in each reference cited is not corrected).

Finally, my revision is not complete as the images of the manuscript are not viewable, although this lack was not a big problem for understanding the manuscript. Therefore, I will wait for the revised version of the manuscript to establish the final outcome of my revision.

Round 2

Reviewer 2 Report

The authors have fully addressed all concerns suggested. In my opinion, the revised manuscript is now suitable for publication in present form, including English language and style.